# Influence of emotions on clinical performance in acute care: A scoping review

Cheng En Xi[1], Sylvain Boet[2,3], Alexandre Assi[4], Lindsey Sikora[5], Meghan M. McConnell[6,7] *

1 Faculty of Medicine, University of Ottawa, Ottawa, Canada, 2 Diving and hyperbaric Unit, Division of Emergency Medicine, Department of Anesthesiology, Clinical Pharmacology, Intensive Care and Emergency Medicine, Geneva University Hospitals and Faculty of Medicine, University of Geneva, Geneva, Switzerland, 3 Ottawa Hospital Research Institute, The Ottawa Hospital, University of Ottawa, Ottawa, Canada, 4 Trinity College Dublin, School of Medicine, Dublin, Ireland, 5 Health Sciences Library, University of Ottawa, Ottawa, Canada, 6 Department of Innovation in Medical Education, University of Ottawa, Ottawa, Canada, 7 Department of Anesthesiology and Pain Medicine, University of Ottawa, Ottawa, Canada

These authors contributed equally to this work.
* meghan.mcconnell@uottawa.ca

## Abstract

Acute care is a high stake, emotionally charged environment. Although emotions are increasingly recognized as integral to various aspects of healthcare, research examining how they influence and interact with clinical performance in acute care settings remains relatively limited. This scoping review aims to summarize relevant empirical research on the influence of emotions on clinical performance in acute care settings. The following databases were searched by a health sciences librarian: Medline and Medline in Process, Embase Classic and Embase, Cochrane's CENTRAL, APA PsycINFO, CINAHL, and ERIC, from inception to June 2024. Empirical research in English related to the effect of emotions on clinical performance in acute care settings were included. The screening was conducted in duplicate independently, and data extraction was done by the lead author and reviewed by a second author. Among 6430 references assessed, 22 studies were analyzed. Three themes were identified based on the research setting: simulated/educational acute care settings, real-world acute care settings, and end-of-life care settings. Overall, negative emotions, most commonly stress, were inversely correlated with clinical performance in some simulated or educational settings and discouraged patient contact in real clinical settings, while positive emotions encouraged more comprehensive care. Experiencing fear and uncertainty led to more cautious care decisions, and negative emotions associated with patient's families were prevalent in end-of-life care. Emotions had varying effects on clinical performance and decision-making in acute care settings, depending on the types of emotions and the clinical contexts. More research is needed to find strategies to help clinicians manage those emotions.

**Data availability statement:** All relevant data are within the manuscript and its Supporting Information files.

**Funding:** The author(s) received no specific funding for this work.

**Competing interests:** The authors have declared that no competing interests exist.

## Introduction

Emotions are abundant within clinical settings. Generally speaking, emotions refer to affective contents, states, and experiences [1], such as fear of appearing incompetent, compassion for patients' suffering, and frustration over lack of institutional resources. Historically, such emotions were viewed in opposition to logical, evidence-based paradigms that dominated medicine. However, researchers have recently started to recognize the importance of emotions in clinical settings [1–3]. For example, Isbell and colleagues [4] described a range of emotions commonly experienced in the emergency department and found that such emotions are the result of a variety of patient (e.g., abusive behaviors), hospital (e.g., limited resources), and system-level (over-crowding) factors. Other research has focused on describing physicians affective responses during emotionally charged events, such as the death of a patient [5], delivering bad news [6], and patient complications [7].

Given the prevalence of emotions within clinical care, researchers have started to investigate the extent to which emotions impact clinical performance. For example, emotions have been shown to influence the quality of doctor-patient interactions [8], clinicians' assessment of risk [9], self-identified medical errors [10], and diagnostic accuracy [11]. The extent to which such emotions influence clinical performance depends on a variety of factors, such as physician personality characteristics, work environment and training background [12]. The impact of emotions on clinical performance also depends on their relevance to the task at hand—specifically, whether the emotions are *integral*, arising directly from the clinical task itself (e.g., stress triggered by a patient's sudden cardiac arrest), or *incidental*, originating from unrelated factors (e.g., irritation caused by loud background noise during a resuscitation) [13]. LeBlanc [14] argued that integral emotions can enhance performance by directing attention toward the task at hand, while incidental emotions may impair performance by diverting attention away from the clinical focus.

Acute care settings—such as emergency departments, intensive care units, and trauma bays— are high-stakes clinical environments characterized by time-sensitive decision-making, diagnostic uncertainty, and interdisciplinary collaboration [15]. These settings provide a unique context for studying the relationship between emotions and clinical performance, which refers to the application of both procedural (e.g., intubation, administering medications) and non-procedural (e.g., decision-making, communication, teamwork) skills during patient care. For instance, "door-to-needle" time in patients with acute myocardial infarction is a well-established prognostic marker, and the ability to meet this benchmark depends on the timely integration of technical expertise, situational awareness, and coordinated team actions [16]. Meanwhile, healthcare professionals report a wide range of emotions, such as stress, fear and frustration [17,18], all of which can affect their clinical performance. Because acute care involves interdisciplinary teams of healthcare professionals [19], each with varying levels of emotional skills training [20], it offers an ideal context for exploring how different specialties perceive, process, and respond to emotions in clinical settings. As such, acute care offers a particularly valuable setting for examining the influence of emotions on clinical performance across a diverse range of situations and professional roles.

This review summarized relevant empirical research examining the influence of emotions on clinical performance in acute care settings. More specifically, our research question was: What is known from existing empirical literature on the influence of emotions on clinical performance in acute care settings?

## Methods

A scoping review was undertaken to systematically review the current landscape of research examining the impact of emotions on clinical performance within the context of acute care. Scoping reviews provide an overview of existing literature in order to identify and map available evidence within a given field [21]. The scoping review methodology was adapted from the JBI Scoping Review Methodology Group [22] framework, and data were reported following the Preferred Reporting Items for Systematic reviews and Meta-Analyses extension for Scoping Reviews (PRISM-ScR) reporting standards [23].

### Search and information sources

The following databases were searched by a health sciences librarian (LS): Medline and Medline in Process via Ovid, Embase Classic + Embase via Ovid, Cochrane's CENTRAL via Ovid, APA PsycINFO via Ovid, CINAHL via EBSCOHost, and ERIC via Ovid. A search strategy was developed in Medline, and then translated into the other databases, as appropriate (see Supplementary Materials). All databases were searched from inception until March 1, 2023, and then updated on June 30, 2024. Additionally, we hand-searched the reference lists of included studies to identify any studies that may have been missed during the search process. We also conducted a forward-citation analysis, which involved using Google Scholar to identify studies that cited these articles.

The screening was limited to empirical research studies published in English. We excluded editorials, letters to the editor, and commentaries. We were interested in studies examining the relationship between emotions and clinical performance in acute care settings, specifically emergency medicine, trauma care, pre-hospital emergency care, acute care surgery, critical care, urgent care, and short-term inpatient stabilization. Accordingly, articles were excluded if the study 1) took place in clinical contexts outside acute care settings, 2) was not specific to healthcare professionals, or 3) focused on psychiatric disorders such as depression and generalized anxiety.

### Study screening

Prior to starting the screening process, a calibration exercise was conducted to ensure reliability in selecting articles. During this exercise, two team members (MM and AA) independently screened a random sample of 5% of the citations to ensure a minimum of 90% inter-rater agreement in screening articles for eligibility or exclusion. Following the calibration exercise, a two-stage screening process took place. During the first stage, titles and abstracts were screened independently and in duplicate by three team members (MM, AA, CX) to identify articles that fit the predefined inclusion criteria. Inter-rater agreement for this initial screen stage was good ($\kappa = 0.74$) [24]. During the second stage of screening, full text articles were reviewed independently by the same three team members. The interrater reliability for this second stage was good ($\kappa = 0.85$) [24]. During both screening phases, the reviewers met regularly to compare their results and discuss any discrepancies.

### Data extraction

Data were extracted from included studies by the lead author (CX) and reviewed by a second author (MMM). Data was extracted using a standardized data extraction sheet that was developed and refined by the research team. Extracted data included basic study details (i.e., author, year, country, and journal), study purpose, study design (i.e., quantitative, qualitative, or mixed methods), definition of emotions, participant characteristics (i.e., sample size, healthcare discipline,

training level), clinical setting, clinical performance outcomes, key study findings, and study limitations. Risk of bias was not examined, as scoping reviews do not typically perform risk of bias analyses.

## Data synthesis

Data were synthesized using both quantitative and qualitative analytical procedures. Quantitative analyses included descriptive frequencies of general characteristics of reviewed research articles. For qualitative analyses, we used a narrative synthesis approach to collate findings and describe patterns across studies. Specifically, each article was reviewed independently by two authors (MMM and CX), and the main findings were extracted and entered in a spreadsheet. We then used inductive coding to translate these findings into themes across studies. The review team met regularly to discuss emergent themes and to derive consensus. Emerging categories and themes were discussed with members of the research team, resulting in three focal themes and subcategories.

## Results

Twenty-two articles met eligibility criteria after screening 6430 references. The screening process and reasons for exclusion are displayed in Fig 1. Majority of the studies were from North America (41%) or Europe (27%), followed by the United Kingdom (18%), with the remainder from Asia (9%) or Oceania (5%). The studies included several different types of healthcare workers/trainees: practicing physicians (n = 12 studies), medical trainees (n = 9), practicing nurses (n = 6), and other allied healthcare professions (n = 4). Six studies examined more than one healthcare professional population.

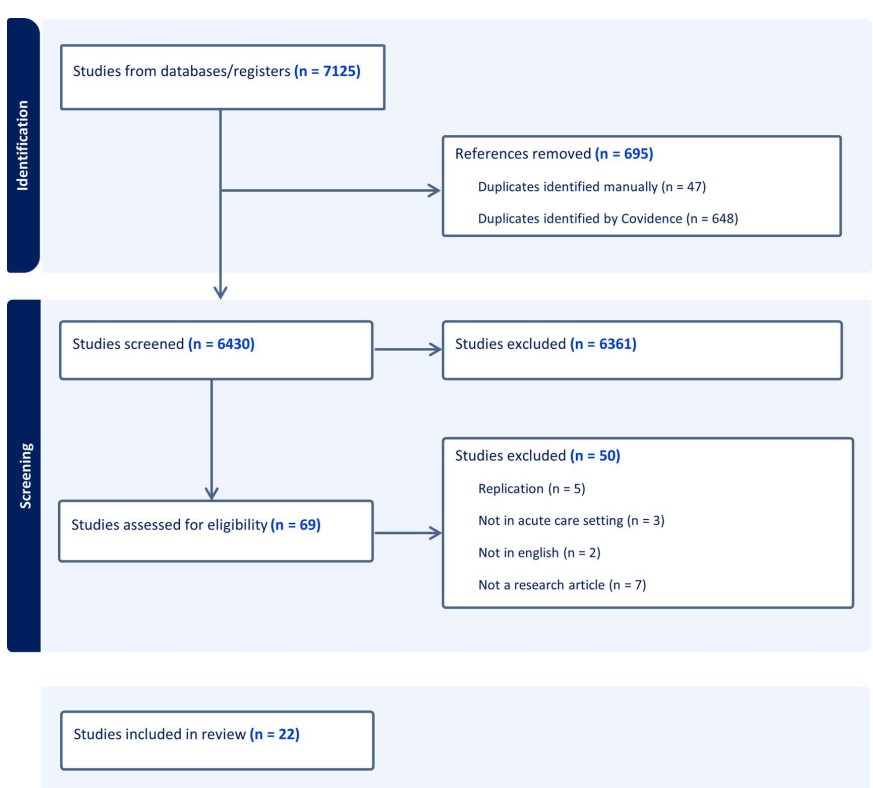

**Fig 1. PRISMA flow chart.**

All the articles were published in or after 2005, with ten studies published since 2019. Thirteen studies were quantitative, eight were qualitative, and one study used a mixed methods design. Five studies did not explicitly examine emotions within their study objectives. For the studies that examined emotions explicitly, the qualitative studies did not operationalize specific emotions but rather kept the definition broad and allowed the participants to discuss the topic more freely in interviews. In contrast, most of the quantitative studies examined negative emotions, with stress and anxiety being the most predominant, with only three studies having assessed positive emotions. More detailed descriptions of the included studies can be found in Table 1.

## Impact of emotions on clinical performance

Three themes were identified across the studies based on the settings they took place: simulated/educational acute care settings, real-world acute care settings, and end-of-life care settings.

### The influence of emotions on clinical skills in simulated/educational acute care settings

Nine studies focused on the effect of emotions on clinical performance in simulated or educational settings. These settings typically involved trainees and assessed both non-procedural (e.g., forming diagnoses) and procedural skills (e.g., lumbar puncture). In five of these studies, negative emotions (e.g., stress, anxiety) were associated with poorer clinical performance [32–36]. For example, LeBlanc et al. [32] randomized paramedic trainees to high- versus low-stress conditions and found that those in the low-stress condition calculated drug doses more accurately than those in the high-stress group. Other studies reported similar findings for non-procedural skills, such as teamwork, task management, and decision-making. For example, in a randomized cross-over trial, Krage et al. [33] assessed anesthesiology residents and staff physicians' procedural and non-procedural skills during two simulated cardiopulmonary resuscitation (CPR) scenarios, with external stressors being present in one of these scenarios. The results showed that non-procedural skills scores were significantly lower when external stressors were present. Moreover, when stressors were present, there was a significant association between non-procedural and procedural skills, indicating that in the presence of stressors, non-procedural skills become increasingly important to maintain procedural CPR performance [33].

Four studies failed to find a relationship between emotions and performance in simulated contexts [36–38]. For example, Geeraerts et al. [37] found no relationship between residents' self-perceived stress and their performance during simulated acute care scenarios. Likewise, Oriot et al. [38] found no significant correlation between medical residents' perceived stress and the success rate of both simulated and real lumbar puncture in infants. Finally, Heyhoe [36] examined the impact of positive and negative affect on the thoroughness of diagnostic information gathering in both medical students (Study 2) and physicians (Study 3), and found no significant differences between affect and diagnostic information gathering.

### The effect of emotions on clinician decision-making in clinical settings

Ten studies took place exclusively in hospital-based acute care settings. Three subthemes emerged from these studies: impact of emotions on quality of care, effects of stress on non-procedural skills, and emotions in the face of uncertainty.

Impact of emotions on quality of care.  Several studies reported that emotions impacted on the quality of care provided by medical staff. Two studies conducted by Isbell and colleagues [4,39] examined emergency department (ED) providers' perceptions of how their emotions impacted their clinical decision-making and patient care. These studies found that clinicians' emotional responses to patients impacted patient engagement and diagnostic decisions [4,39]. For example, patients with unrealistic expectations led to frustration and anger among ED staff, which was then associated with poorer quality care and less time spent with patients (e.g., less frequent room visits). In contrast, kind and appreciative patients inspired clinicians to spend more time with them and go the extra mile [4]. Relatedly, negative emotions elicited by systemic factors also negatively impacted patient engagement. For example, Safi-Keykaleh et al. [40] interviewed Iranian

**Table 1. Overview of the studies included.**

| First author, year | Study design | Healthcare Profession and sample size | Healthcare Unit | Clinical performance | Emotional factors investigated | Emotion measurement methods (if applicable) |
|---|---|---|---|---|---|---|
| Arnetz et al, 2017 | Quantitative: Cross-sectional | Postgraduate Medical Trainees (n = 28) | Emergency department | Number of near misses when working with critically ill and trauma patients. | Stress | Self-reported measure: Single item numerical scale, ranging from 0–10 Physiological stress markers: blood and saliva samples |
| Carenzo et al, 2020 | Quantitative: Cross-sectional | Postgraduate Medical Trainees (n = 95) | Simulated medical emergency scenario competition. | Procedural and non-procedural skill proficiency | Challenge and threat states, cognitive anxiety, and somatic anxiety | Challenge and threat appraisals: Demands and Resources Evaluation Scale (DRES) [25] Cognitive anxiety and somatic anxiety: subscales of Mental Readiness Form (MRF) [26] |
| De Boer et al, 2022 | Qualitative: Interpretive Description | Physicians (n-4), Nurses (n = 5) | Critical care/ICU | End-of-life (EOL) decision-making | All emotions were explored during the interviews. | Not applicable |
| Geeraerts et al, 2017 | Quantitative: Correlational/observational study | Postgraduate Medical Trainees (n = 27) | Critical care/ICU; Anesthesiology | Procedural skills: the time needed to understand and correctly identify the cause of the crisis, implement correct measures, and a global rating of the quality of care. Non-procedural skills: task management, team working, situation awareness, decision making. | Stress | Self-reported measure: Single item numerical scale, ranging from 0–10 Physiological stress marker: salivary amylase concentration. |
| Heyhoe, 2013, Study 2 | Quantitative: Experimental | Undergraduate Medical Trainees (n = 93) | Acute care hospital | Diagnostic stage of information gathering, more specifically the appropriateness of important facts chosen for diagnosis, and thoroughness of information gathering | Mood (valence and arousal) Anticipatory affect Anticipated affect | Valence: Single item numerical scale, ranging from 0–7 Arousal: Single item numerical scale, ranging from 0–7 Anticipatory affect: Positive and Negative Affect Schedule – Expanded Form (PANAS-X) [27] Anticipated affect: Six item Likert-scale, with 7-point rating scale |
| Heyhoe, 2013, Study 3 | Quantitative: Experimental | Physicians (n = 77) | Acute care hospital | Thoroughness and sequence in which doctors gather clinically relevant information may have important implications for whether a doctor will be on the right trajectory for making an accurate diagnosis | Mood (valence and arousal) Anticipatory affect Anticipated affect | Valence: Single item numerical scale, ranging from 0–7 Arousal: Single item numerical scale, ranging from 0–7 Anticipatory affect: Positive and Negative Affect Schedule – Expanded Form (PANAS-X) [27] Anticipated affect: Six item Likert-scale, with 7-point rating scale |
| Heyhoe, 2013, Study 4 | Quantitative: Cross-sectional | Physicians (n = 27), Nurses (n = 25), Physiotherapists (n = 2) | Acute care hospital | Individual and team communication behaviour and team effectiveness | Individual and team affect. | Individual affect: International Positive and Negative Affect Schedule (I-PANAS-SF) [28] Team affect: Two item Likert-scale, with a 5-point scale |
| Heyhoe, 2013, Study 5 | Qualitative: Critical Decision Method | Physicians (n = 16) | Emergency department; Anesthesiology | Emergency care decision making | All emotions were explored during the interviews. | Not applicable |
| Hwang & Shin, 2023 | Quantitative: Cross-sectional | Nurses (n = 156) | Emergency department | Triage reasoning competence: the ability to triage patients effectively. | Workplace stress | Korean Nursing Stress Scale [29] |

*(Continued)*

**Table 1.** (Continued)

| First author, year | Study design | Healthcare Profession and sample size | Healthcare Unit | Clinical performance | Emotional factors investigated | Emotion measurement methods (if applicable) |
|---|---|---|---|---|---|---|
| Isbell, Boudreaux, et al, 2020 | Mixed Methods: Parallel Design | Physicians (n=50), Nurses (n=44) | Emergency department | Clinical decision-making. | Specific emotions associated with patient, hospital, and system-level factors | Not applicable |
| Isbell, Tager, et al, 2020 | Qualitative: Grounded theory | Physicians (n=45), Nurses (n=41) | Emergency department | Clinical decision making, overall care quality, and patient safety. | Emotions associated with positive and negative patient encounters, as well as emotions elicited by mental health patients. | Not applicable |
| Katz et al 2005 | Quantitative: Cross-sectional | Physicians (n=33) | Emergency department | Triage decisions and diagnostic testing of patients with possible acute cardiac ischemia | Fear of malpractice. | Six item Likert-scale, with 5-point rating scale |
| Kohut et al, 2020 | Qualitative: Not stated | Physicians (n=18), Nurses (n=3), Physician assistant (n=4) | Urgent care | Prescription of antibiotics | Emotions were not part of the study objective. | Not applicable |
| Krage et al, 2017 | Quantitative: Experimental | Postgraduate Medical Trainees (n=20), Physicians (n=10) | Simulated cardiopulmonary resuscitation | 1) Non-procedural skills, such as task management, teamwork, situation awareness, and decision-making. 2) Procedural skills while performing CPR. | Stress | Stress was not measured |
| LeBlanc et al, 2005 | Quantitative: Experimental | Paramedics (n=30) | Pre-hospital emergency care | Drug dosage calculations | Stress | State-Trait Anxiety Inventory (STAI) [30] |
| Mehter et al, 2018 | Qualitative: Grounded theory | Physicians (n=18) | Critical care/ICU | End-of-life decision-making. | Emotions were not part of the study objective. | Not applicable |
| Oriot et al, 2021 | Quantitative: Cross-sectional | Postgraduate Medical Trainees (n=33) | Pediatric emergency department | Lumbar puncture performance and success rates in simulated and clinical settings. | Stress | Self-reported measure: Single item numerical scale, ranging from 0–10 Physiological stress marker: salivary cortisol. |
| Pottier et al, 2013 | Quantitative: Experimental | Undergraduate Medical Trainees (n=41) | Simulated ambulatory consultation | Diagnostic accuracy and communication skills. | Stress | Self-report measures: Single item numerical scale, ranging from 0–100; French version of the State-Trait Anxiety Inventory (STAI) [31] Physiological stress marker: salivary cortisol. |
| Robertsen et al, 2019 | Qualitative: Not stated | Physicians (n=18) | Trauma care center | Decisions about whether or not to withdraw life sustaining treatment after devastating brain injury (DBI) | Doubt | Not applicable |
| Rotella et al, 2014 | Quantitative: Survey | Undergraduate Medical Trainees (n=23), Postgraduate Medical Trainees (n=26) | Acute care hospital | Junior medical officers decisions to escalate patient care to a senior colleague | Emotions were not part of the study objective. | Not applicable |

*(Continued)*

**Table 1.** (Continued)

| First author, year | Study design | Healthcare Profession and sample size | Healthcare Unit | Clinical performance | Emotional factors investigated | Emotion measurement methods (if applicable) |
| --- | --- | --- | --- | --- | --- | --- |
| Safi-Keykaleh et al, 2022 | Qualitative: Grounded theory | Emergency medicine technicians (n = 26) | Pre-hospital emergency care | On scene decision making by EMTs, starting from the scene until the patient is delivered to the hospital | Emotions were not part of the study objective. | Not applicable |
| Schoenfeld et al, 2019 | Qualitative: Not stated | Physicians (n = 15) | Emergency department | Shared decision making between healthcare providers and patients. | Emotions were not part of the study objective. | Not applicable |

emergency medical technicians (EMTs), and found that systemic barriers, such as unclear delineation of responsibilities and poor liability coverage, led to stress and frustration. These negative emotions led EMTs to avoid making medical decisions during patient transport, jeopardizing patient safety. Lastly, Heyhoe (Study 5) [36] interviewed emergency and anesthesia physicians to identify emotional processes that influence clinical decision-making in emergency care. Positive affect (e.g., calm, comfort) was thought to improve clinical judgment by facilitating a more focused approach to clinical judgments. In contrast, negative affect (e.g., pressure, stress, fear) was regarded as a barrier to timely decision-making.

Effects of stress on non-procedural skills. Two studies examined the relationship between clinician' self-reported stress and non-procedural skills in acute care settings, and reported contrasting findings [41,42]. Specifically, Arnetz et al. found a positive correlation between the stress levels of emergency medicine residents and their number of "near misses" in trauma settings [41]. However, Hwang and Shin found that higher levels of stress among emergency department nurses was associated with better triage competence [42].

Dealing with fear and uncertainty in clinical contexts. Lastly, several studies reported that clinicians made more cautious clinical decisions when faced with fear and uncertainty [43–46]. For example, Katz et al. [43] found that physicians with a greater fear for malpractice were more likely to admit a patient for suspected Acute Cardiac Ischemia, and were more likely to order more diagnostic tests, compared to physicians with less malpractice fear. Kohut et al. [44] found that both diagnostic uncertainty and fear of missing a bacterial infection increased unnecessary antibiotic prescriptions. Rotella et al. [45] found that clinical uncertainty was an important determinant of residents' decision to escalate patient care to senior colleagues. Lastly, Schoenfeld and colleagues [46] explored barriers and facilitators of shared decision-making (SDM) in the ED. Emergency physicians described uncertainty as an emotional challenge that often impacted SDM negatively. Specifically, physicians described how fear of a bad outcome acted as a barrier to SDM; for example, involving patients in decision-making could result in a more 'conservative' decision, which could lead to a missed diagnosis.

## Emotions and unspecified care decisions in end-of-life care

Three qualitative studies assessed clinician emotions during end-of-life (EOL) care in critical care settings [47–49]. Two studies specifically assessed the effect of emotions on EOL decision-making [47,49], while one study examined physicians' approach EOL care more broadly, with emotional factors being a finding [48]. Prevailing findings were the association of family conflict with negative emotions, and the desire to act in the patients' best interests and avoid futile care. For instance, in De Boer et al., [47] physicians and nurses reported that unreceptive family members led to negative emotions, such as stress, annoyance, and frustration. Conflict with family members was often associated with the decision to withdraw futile care, whereby physicians described their desire to act in the best interest of the patient to minimize suffering, with the patients' family members not always receptive to these decisions. Physicians and nurses stated such emotions affected their EOL decision-making, but the study did not specify the effects [47]. The effects of conflict with families were

more evident in the study by Mehter et al. [48], whereby physicians reported feeling anxious when anticipating conflict with patients' families. This anxiety caused physicians to avoid or postpone meetings with families. Interestingly, some patient populations are more likely to trigger emotions among physicians. In Robertsen et al., physicians reported that emotions were a major factor when working with pediatric patients, as they reminded them of their own children, which influenced their clinical judgment [49]. However, the study did not specify the type of emotions nor the direction of the influence.

## Discussion

This scoping review found that negative emotions were inversely correlated with clinical performance in some simulated or educational settings and discouraged patient contact in real clinical settings, while positive emotions encouraged more comprehensive care. Additionally, experiencing fear and uncertainty led to more cautious care decisions, and negative emotions associated with patient's families (e.g., stress, frustration) were prevalent in EOL care. Despite the importance of emotions on patient safety [50], little research has been done on their effects in acute clinical practice. Most research on this topic has centered around identifying the presence of negative emotions, most commonly stress, among healthcare workers, and elucidating effects of emotion on well-being and patient care. However, compared to other high-risk industries, such as the military [51], there is a lack of research focused on identifying effective emotion regulation strategies and integrating them into clinical education or training programs. Therefore, given this review's findings, future research should explore emotion management strategies to maximize well-being and patient safety.

Regarding simulation/educational settings, clinical performance was particularly affected by negative incidental emotions—those triggered by external stressors unrelated to the clinical task—compared to integral emotions, which arise directly from the task itself. In the context of the present findings, those studies found that a significant effect of emotions on clinical performance focused more on incidental emotions as opposed to integral emotions. For example, three of the five studies that found an effect of emotions on clinical performance were experimental designs, whereby high-stress conditions were compared against low-stress conditions. In these studies, high stress was experimentally manipulated by the addition of external stressors, such as an unrelated challenging task [32], aggressive family members [33,34], and/or external noise [33]. These unrelated stressors induced incidental affective states. The remaining two studies did not explicitly induce negative moods through external stressors, but rather, measured healthcare professional's self-reported affective states either before [35] or after (Heyhoe, Study 4) [36] participation in a series of simulated scenarios. In this way, the focus was on affective states that were not directly associated with (e.g., were incidental to) the clinical task.

In contrast, three of the four studies that failed to find a relationship between emotions and clinical performance focused more on integral emotions. These studies measured self-perceived stress associated with the clinical task, without experimental conditions that would reasonably create incidental emotions [36,38]. However, Geeraerts et al. [37] did not explicitly state whether there were conditions that would create incidental emotions. The researchers only measured stress before and after acute care scenarios, and they failed to find an effect of stress on clinical performance. This finding suggests that factors beyond the distinction between integral and incidental emotions may contribute to the observed outcomes. Nonetheless, it is possible that incidental emotions exert a greater influence on clinical performance in simulated settings, whereas negative integral emotions may have a comparatively limited impact in these controlled environments. There is little research examining the effects incidental and integral emotions on clinical performance, particularly within clinical contexts. Acute care settings are rich with both integral (e.g., unanticipated difficult airway) and incidental (e.g., noise, difficult colleagues) emotions. Simulation provides us with the opportunity to study the impact of both types of emotions on clinical performance, and future research should attempt to differentiate between their effects on acute care clinical performance.

In addition to examining the impact of emotions on clinical performance in simulated or educational settings, our study also highlighted the impact of emotions in hospital-based acute care settings. Our findings indicate that positive and negative emotions may differentially impact clinical performance. Generally speaking, healthcare providers report

negative emotions, such as stress and frustration, in response to both patient encounters [4,36,39] and systemic factors [4,33]. These emotions are thought to adversely affect patient engagement and clinical decision-making, leading to lower quality patient care. In contrast, positive emotions, such as kindness and appreciation, are associated with greater patient engagement and believed to enhance patient care [4,36,39]. While healthcare providers expressed awareness that negative emotions can adversely impact patient care [4,36,39], there was little discussion of clinicians' efforts to change such behaviors to ensure equitable care to all patients. This observation potentially indicates a lack of training on how to work with more difficult patients. There is evidence to suggest that providing trainees with opportunities to interact with difficult and/or challenging patients in simulated contexts can improve confidence and communication competencies [52,53]. Future research should examine whether exposure to emotionally evocative scenarios in simulated contexts helps trainees deal effectively with difficult patients in real clinical practice.

In terms of non-procedural skills in real-world clinical settings, our results were inconclusive [41,42]. Hwang and Shin [42] found that stress positively influenced nurse triage performance, but this effect was mediated by effective nurse-physician collaboration. In contrast, Arnetz et al. [41] found that high levels of self-reported stress were associated with the number of "near misses", as estimated by their attending physician. The methodological differences between the two studies make it difficult to compare the results. Additionally, the two studies were conducted in different countries: South Korea and the US. Cultural and systemic factors may have also influenced how clinicians responded to stressful environments. Nonetheless, given the small number of studies within this category, more research is needed to elucidate the impact of emotions on non-procedural skills in acute care settings.

Another key finding from our scoping review was that fear and clinical uncertainty was associated with more cautious decision-making in acute care settings. In some cases, this cautious decision-making had positive implications. For example, in the study be Rotella et al. [45], diagnostic uncertainty led residents to escalate care to their supervisors despite their fear of criticism. However, in other cases, such decision-making may have unintended negative consequences. In the study by Kohut et al. [44], diagnostic uncertainty surrounding a potential bacterial infection contributed to the overuse of antibiotics. Given the prevalence of uncertainty in the practice of medicine, a growing body of research has explored its impact within acute care contexts. Intolerance to uncertainty has been associated with increased ordering of tests, reduced comfort with more complex patients, and delayed care [54,55]. Substantially less research has examined how emotional responses to uncertainty may mediate its effects on clinical performance. Findings from this review suggest that such emotional responses – particularly fear – can impact various clinical outcomes, highlighting the need for future research in this area.

In the context of end-of-life care, our scoping review revealed a lack of clear connections between emotional states and specific clinical performance outcomes. The studies reviewed did not identify or define the types of emotions experienced by healthcare professionals (e.g., stress, guilt, compassion, or joy), nor did they specify the clinical decisions influenced by these emotional states. Instead, findings were generally limited to broad descriptions of a desire to minimize patient suffering and adhere to best practices or evidence-based guidelines. As a result, the specific role emotions play in shaping patient care in end-of-life contexts remains underexplored, highlighting a critical gap and an important direction for future research. Although some studies referenced specific emotions, such as doubt, uncertainty, and anxiety [47,49], these were not directly linked to clinical performance. Additionally, conflict with patients' family members regarding goals of care emerged as a common emotional trigger across three studies [47–49]. Future research should aim to identify the specific emotions elicited by such conflicts and examine how these emotional experiences influence care decisions and clinical performance in end-of-life settings. This remains an important and underexplored area with significant implications for both practice and training.

Our findings have numerous practical implications for clinical care and medical education. For one, medical educators should account for incidental emotions during clinical training. While simulation training can adequately reflect integral emotions in the real world [38], there needs to be more training centered around realistic clinical distractors that could

elicit negative incidental emotions. For example, Feuerbacher et al. [56] employed four common clinical distractors during a cholecystectomy simulation and found that the majority of errors occurred while residents were exposed to these distractions. Incorporating such training could help trainees become more accustomed to such distractors and develop coping strategies for managing them in real-world clinical settings.

Furthermore, the neglect of more difficult patients has the potential to exacerbate existing health inequities. In some cases, difficult patients are a result of social inequalities that lead to conflict with physicians, such as treatment noncompliance due to low income [57]. The inverse care law, an observation that the most disadvantaged individuals tend to face the most access barriers [58], could be applied to the effects of emotions on acute care. Clinicians experience negative emotions with difficult patients, many of whom may be socioeconomically disadvantaged, and provide lower-quality care because of the emotions, further exacerbating existing inequalities to those patients who may need more care. More research in this area and finding effective solutions are crucial to reducing existing health inequalities.

Regarding the issue of uncertainty, there are implications for antibiotics stewardship and health resource conservation. Antibiotics overuse accelerates antimicrobial resistance [59] and leads to an increased risk of adverse effects and over-medicalization of self-limiting conditions. Meanwhile, acute care beds are in short supply [60], and extended hospital stays and admissions of patients who may not require that level of care further strain the system. A better understanding of the emotional effect of uncertainty on clinical care and effective strategies to manage such emotions could help increase resource use efficiency.

While most studies have focused on negative emotional states, emerging literature suggests that positive emotions may also enhance clinical performance. For example, medical students induced into a positive emotional state diagnosed patient cases with the same level of accuracy but significantly faster than those in a neutral emotional state [61]. Similarly, practicing anesthesiologists who reported experiencing more frequent positive emotions initiated airway management interventions more quickly during a simulated scenario [62]. These findings highlight the potential value of further exploring the role of positive emotions in clinical performance and identifying strategies to leverage these emotional states to improve patient outcomes.

While scoping reviews do not require formal quality appraisal, it is worth noting several methodological considerations across the included studies. The majority employed quantitative methodologies, most commonly experimental or cross-sectional designs. Experimental studies often relied on external raters to evaluate clinical performance using standardized numerical rating scales. While this design improves internal validity through controlled conditions, such control may reduce ecological validity and limit generalizability to real-world clinical settings. Cross-sectional studies, frequently relying on self-reported emotional states, are particularly susceptible to biases such as social desirability, and often do not account for causal or mediating relationships between emotions and performance. Several studies used qualitative methods, which allowed for a richer, more nuanced understanding of emotional and clinical experiences. However, qualitative findings are typically not intended to generalize outside the study context, and nearly half of the qualitative studies in this review did not specify the methodological framework guiding data collection and analysis. Most notably, across both qualitative and quantitative studies, there was a consistent lack of theoretical or conceptual clarity regarding emotional constructs. Many studies failed to define specific emotions, even when referencing commonly studied states such as stress, or used vague terms such as "positive" or "negative" emotions without justification for dimensional approaches. Future research in this area would benefit from greater methodological transparency and the explicit definition and theoretical grounding of emotional constructs under investigation.

## Limitations

While this review provides insights into the influence of emotions on clinical performance in acute care contexts, the results should be viewed in the context of study limitations. First, while we adhered to a strict study selection and scoping

review methodology, we cannot rule out the possibility that relevant studies may have been missed. Second, while we tried to mitigate subjectivity by using at least two members of the research team during data collection, extraction, and syntheses, our own biases and conceptualization of emotions may have impacted how we viewed and analyzed the data. Third, our search strategy was limited to articles published in English, so we may have overlooked relevant non-English articles. Moreover, most (89.4%) of these articles originated in Western settings, which limits the generalizability of the findings, and highlights the need for cross-cultural studies to examine possible cultural variations in how emotions impact clinical performance in acute care settings. Fourth, as is common in scoping reviews [22], we did not conduct a critical appraisal or risk of bias assessment, and therefore we cannot comment on the methodological quality of the individual studies. Lastly, there are variety of affective constructs, such as emotional intelligence and motivation, that likely influence clinical performance in acute care settings; these were outside the scope of this review but provide areas for future research.

## Conclusions

Clinicians routinely experience a wide range of emotions in acute care settings, and these emotional states can significantly influence clinical performance, depending on the nature of the emotion and the clinical context in which it arises. This scoping review identified evidence linking emotions to both procedural and non-procedural performance, as well as to decision-making under conditions of fear and uncertainty. Collectively, these findings underscore the need for greater attention to the emotional dimensions of acute care clinical work—both in research and in the design of educational interventions aimed at supporting clinicians' performance and well-being in high-stakes environments.

## Supporting information

**S1 Appendix. Key words used in the search strategy.**
(DOCX)

**S1 Data. Data Extraction.**
(ZIP)

## Acknowledgments

None.

## Author contributions

**Conceptualization:** Sylvain Boet, Meghan McConnell.

**Data curation:** Lindsey Sikora, Meghan McConnell.

**Formal analysis:** Cheng En Xi, Alexandre Assi, Lindsey Sikora, Meghan McConnell.

**Methodology:** Sylvain Boet, Lindsey Sikora, Meghan McConnell.

**Project administration:** Meghan McConnell.

**Resources:** Lindsey Sikora.

**Software:** Lindsey Sikora.

**Supervision:** Meghan McConnell.

**Writing – original draft:** Cheng En Xi, Meghan McConnell.

**Writing – review & editing:** Cheng En Xi, Sylvain Boet, Alexandre Assi, Meghan McConnell.

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
