## [Decision Letter · Decision Letter 0]

PONE-D-25-06681Influence of emotions on clinical performance in acute care: a scoping reviewPLOS ONE

Dear Dr. McConnell,

Thank you for submitting your manuscript to PLOS ONE. After careful consideration, we feel that it has merit but does not fully meet PLOS ONE’s publication criteria as it currently stands. Therefore, we invite you to submit a revised version of the manuscript that addresses the points raised during the review process.

We look forward to receiving your revised manuscript.

Kind regards,

Maheshkumar Baladaniya

Academic Editor

PLOS ONE

**Journal Requirements:**

1. When submitting your revision, we need you to address these additional requirements. Please ensure that your manuscript meets PLOS ONE's style requirements, including those for file naming. The PLOS ONE style templates can be found at https://journals.plos.org/plosone/s/file?id=wjVg/PLOSOne_formatting_sample_main_body.pdf and https://journals.plos.org/plosone/s/file?id=ba62/PLOSOne_formatting_sample_title_authors_affiliations.pdf

**Additional Editor Comments:**

Some minor changes should be conducted from the author side.

Reviewers' comments:

Reviewer's Responses to Questions

**Comments to the Author**

1. Is the manuscript technically sound, and do the data support the conclusions?

Reviewer #1: Yes

Reviewer #2: Yes

2. Has the statistical analysis been performed appropriately and rigorously? 

Reviewer #1: Yes

Reviewer #2: I Don't Know

3. Have the authors made all data underlying the findings in their manuscript fully available?

Reviewer #1: Yes

Reviewer #2: Yes

4. Is the manuscript presented in an intelligible fashion and written in standard English?

Reviewer #1: Yes

Reviewer #2: Yes

5. Review Comments to the Author

**Reviewer #1: ** This manuscript provides a well-executed scoping review examining how emotions influence clinical performance in acute care settings. It addresses a significant and understudied issue in healthcare—how affective states (both positive and negative) shape clinical decision-making, skill performance, and provider behavior. The manuscript follows appropriate scoping review methodology, reports data clearly, and draws meaningful implications for clinical training and future research.

Comments:

The paper sometimes uses terms like “integral” and “incidental” emotions without early definition. These should be clearly defined in the introduction or methods for a broader audience. “Negative emotions” is used as a catch-all term—more nuanced language (e.g., anger, fear, sadness, stress) could improve interpretability in certain sections. Add a table summarizing emotional constructs across studies (e.g., specific emotions examined, definitions used, measurement methods).

While scoping reviews don't typically require a formal quality appraisal, readers would benefit from a brief comment on the general methodological quality of the included studies (e.g., experimental vs. observational vs. qualitative). Include a short paragraph noting the predominant study designs and potential limitations in evidence quality.

Most of the analysis focuses on negative emotional states. While this reflects the literature, the potential benefits of positive emotions (e.g., joy, compassion, gratitude) in clinical performance are underexplored in the discussion. Strengthen discussion on how positive emotions could be leveraged in training or decision-making.

The section on end-of-life care lacks clear connections between emotional states and specific performance outcomes. Acknowledge this more explicitly in the discussion as a gap in the literature and a target for future research.

**Reviewer #2: ** This scoping review offers a well-structured and informative exploration of the existing literature on the influence of emotions on clinical performance in acute care. The authors clearly articulate the purpose of the review, and the research objectives are appropriately aligned with the scoping review methodology. However, for the introduction part, it is suggested that the author to add more explanation on clinical performance and acute care setting.

The use of established frameworks, such as the JBI Scoping Review Methodology Group framework, adds methodological rigor and transparency to the review process. The inclusion and exclusion criteria are well-defined, and the selection process is clearly documented through the flow diagram.

The literature search appears systematic and comprehensive, involving multiple relevant databases and search terms. The data charting and extraction process is described in sufficient detail, and the results are presented in a logical and accessible manner.

One of the key strengths of this review is its ability to synthesize a broad body of literature and identify key themes and gaps, offering valuable insights for future research and practice.

Overall, the manuscript is clearly written, logically organized, and professionally presented.

6. PLOS authors have the option to publish the peer review history of their article (what does this mean? ). If published, this will include your full peer review and any attached files.

**Do you want your identity to be public for this peer review?** For information about this choice, including consent withdrawal, please see our Privacy Policy .

Reviewer #1: No

Reviewer #2: No

---

## [Author Response · Author response to Decision Letter 1]

16 Jun 2025

Reviewer #1 Comments:

1. “The paper sometimes uses terms like “integral” and “incidental” emotions without early definition. These should be clearly defined in the introduction or methods for a broader audience.”

Response: This is a good point, and we previously decided to clarify these terms in the discussion, as we did not anticipate this distinction to come up with the results. We added a section within the Introduction to define integral and incidental emotions and provide examples of how they affect clinical performance differently within the current literature (the paragraph starting line 44. We also provided evidence from LeBlanc et al. that integral emotions can direct attention to a clinical task while incidental emotions can direct attention away.

2. “Negative emotions” is used as a catch-all term—more nuanced language (e.g., anger, fear, sadness, stress) could improve interpretability in certain sections. Add a table summarizing emotional constructs across studies (e.g., specific emotions examined, definitions used, measurement methods).”

Response: Thank you for this comment. Rather than add an entirely new table, we added a column to Table 2 to summarize the emotional constructs across the studies, and the measurement of these constructs (where applicable). We also altered text throughout the manuscript to be more specific, referencing specific emotional states whenever possible. Admittedly, part of the challenge we had is that most studies did not define emotions, or the definitions were not applicable. As such, we also elaborated on the lack of clear definitions of emotional constructs in our Discussion section (lines 472-477).

3. “While scoping reviews don't typically require a formal quality appraisal, readers would benefit from a brief comment on the general methodological quality of the included studies (e.g., experimental vs. observational vs. qualitative). Include a short paragraph noting the predominant study designs and potential limitations in evidence quality.”

Response: Several paragraphs discussing the most predominant methodologies and their strengths and limitations were added just prior to the Limitations section (paragraph starting Line 459). In addition to elaborating on the strengths and limitations of the different types of methodologies used by the studies in our review, we also noted that there was a lack of theoretical or conceptual clarity regarding emotional constructs (see Comment 2, above).

4. “Most of the analysis focuses on negative emotional states. While this reflects the literature, the potential benefits of positive emotions (e.g., joy, compassion, gratitude) in clinical performance are underexplored in the discussion. Strengthen discussion on how positive emotions could be leveraged in training or decision-making.”

Response: A section on the effects of positive emotions on patient outcomes was added in the paragraph that discusses the practical implications of our findings (starting Line 439). We stated that “while most studies have focused on negative emotional states, emerging literature suggests that positive emotions may also enhance clinical performance” and provided two examples. We further stated that “these findings highlight the potential value of further exploring the role of positive emotions in clinical performance and identifying strategies to leverage these emotional states to improve patient outcomes”.

5. “The section on end-of-life care lacks clear connections between emotional states and specific performance outcomes. Acknowledge this more explicitly in the discussion as a gap in the literature and a target for future research.”

Response: The description on this research gap was revised in the paragraph starting Line 376. The wording was changed to more explicitly identify this research gap and the need to address it in future research. We stated that “the studies reviewed did not identify or define the types of emotions experienced by healthcare professionals (e.g., stress, guilt, compassion, or joy), nor did they specify the clinical decisions influenced by these emotional states.” and “future research should aim to identify the specific emotions elicited by such conflicts and examine how these emotional experiences influence care decisions and clinical performance in end-of-life settings.”

Reviewer #2 Comment:

1. “However, for the introduction part, it is suggested that the author to add more explanation on clinical performance and acute care setting.”

Response: This comment has been addressed in the paragraph starting Line 62. First, we provided elaboration on what we mean by “acute care” and “clinical performance”. We added a section that provided more detail on the time-sensitivity of cute care, using the “door-to-needle time” of acute myocardial infarction as an example. Additionally, we stated that “healthcare professionals report a wide range of emotions being present in the acute care setting” and “the acute care setting consists of interdisciplinary healthcare professionals, and the emotion skills training of these professional are heterogenous”. We believe these additional details help contextualize why the acute care setting is uniquely suited to assess the effect of emotions on clinical performance across different healthcare specialties.

---

## [Decision Letter · Decision Letter 1]

Influence of emotions on clinical performance in acute care: a scoping review

PONE-D-25-06681R1

Dear Dr. McConnell,

We’re pleased to inform you that your manuscript has been judged scientifically suitable for publication and will be formally accepted for publication once it meets all outstanding technical requirements.

Kind regards,

Maheshkumar Baladaniya

Academic Editor

PLOS ONE

Additional Editor Comments (optional):

None.

Reviewers' comments:

Reviewer's Responses to Questions

**Comments to the Author**

1. If the authors have adequately addressed your comments raised in a previous round of review and you feel that this manuscript is now acceptable for publication, you may indicate that here to bypass the “Comments to the Author” section, enter your conflict of interest statement in the “Confidential to Editor” section, and submit your "Accept" recommendation.

Reviewer #1: All comments have been addressed

Reviewer #3: All comments have been addressed

2. Is the manuscript technically sound, and do the data support the conclusions?

Reviewer #1: Yes

Reviewer #3: Yes

3. Has the statistical analysis been performed appropriately and rigorously? 

Reviewer #1: Yes

Reviewer #3: Yes

4. Have the authors made all data underlying the findings in their manuscript fully available?

Reviewer #1: Yes

Reviewer #3: Yes

5. Is the manuscript presented in an intelligible fashion and written in standard English?

Reviewer #1: Yes

Reviewer #3: Yes

6. Review Comments to the Author

Reviewer #1: Thank you for revision. All comments are addressed by authors and no further revision is required. Manuscript is ready for further process.

Reviewer #3: (No Response)

7. PLOS authors have the option to publish the peer review history of their article (what does this mean? ). If published, this will include your full peer review and any attached files.

**Do you want your identity to be public for this peer review?** For information about this choice, including consent withdrawal, please see our Privacy Policy .

Reviewer #1: No

Reviewer #3: No
